# The Italian Framework of Bipolar Disorders in the Elderly: Old and Current Issues and New Suggestions for the Geriatric Psycho-Oncology Research

**DOI:** 10.3390/biomedicines11051418

**Published:** 2023-05-11

**Authors:** Vincenza Frisardi, Chiara Pollorsi, Luisa Sambati, Maria Macchiarulo, Andrea Fabbo, Francesca Neviani, Marco Menchetti, Rabih Chattat

**Affiliations:** 1Geriatric Unit Neuromotor Department, AUSL-IRCCS of Reggio Emilia, 80 Risorgimento Avenue Reggio Emilia, 46123 Reggio Emilia, Italy; 2U.O.C. Clinica Neurologica Rete Metropolitana (NeuroMet), IRCCS Istituto delle Scienze Neurologiche di Bologna, 40139 Bologna, Italy; 3Acute Geriatrics and Orthogeriatrics and Cognitive Disorders and Dementia Center, IRCCS AOU-BO, 40138 Bologna, Italy; 4Geriatric Service, Cognitive Disorders and Dementia Unit, Health Authority and Services (AUSL) of Modena, 41124 Modena, Italy; 5Geriatric Unit, Azienda Ospedaliera Universitaria Policlinico di Modena, 41124 Modena, Italy; 6Department of Biomedical and NeuroMotor Sciences (DiBiNeM), University of Bologna, 40127 Bologna, Italy; 7Department of Psychology, University of Bologna, 40127 Bologna, Italy

**Keywords:** bipolar disorders, elderly, psychogeriatric research, psycho-oncogeriatrics

## Abstract

Background: Older adults with mood disorders constitute a heterogeneous group in a complex spectrum interlinked with physical comorbidities. Worldwide, Bipolar disorders in older people (OABD) remain underestimated and underdiagnosed. OABD is challenging in the clinical setting and is associated with adverse outcomes (increased risk of anti-social behaviour triggered by inappropriate drugs and increased incidence of health deficits, including cancer). This article aims to describe the state of the art of OABD in the Italian framework and provide a new field of research. Methods: We performed an overview of the literature, selecting our target population (over 65 years) and synthesising the main challenging issues. By exploiting the Italian database from the Minister of Health in 2021, we analysed epidemiological data in the age range 65–74 years and 75–84 years old. Results: Females showed the highest prevalence and incidence in both groups, with a regional difference across the country but more evident in the Autonomous Provinces of Bolzano and Trento for the 65–74 years range. Several projects recently focused on this topic, and the urgency to define better the epidemiological framework is mandatory. Conclusions: This study represented the first attempt to report the comprehensive Italian framework on OABD aimed at fostering research activities and knowledge.

## 1. Introduction

By 2050, the share of the world’s older adults is estimated to nearly double from 12% to 22%, and over 20% of adults over 60 years old suffer from a mental or neurological disorder [1]. After the COVID-19 pandemic, these estimates could be regarded as an important public health problem, mainly because the stigma surrounding these conditions makes people reluctant to seek help appropriately or due to the lack of healthcare resources. Italy has the third oldest population in the world and the first European country penalised for the COVID-19 pandemic with its implications on the well-being and quality of care of vulnerable patients.

Older adults with mood disorders constitute a heterogeneous group showing a complex spectrum of symptoms ranging from psychological frailty reactions to significant severe depressive or manic episodes. The Diagnostic and Statistical Manual 5 edition (DSM-5) describes bipolar I disorder (BD-I) and bipolar II disorder (BD-II) as the two major subtypes of Bipolar Disorder (BD) [2], while other diagnoses in bipolar-related disorders make the diagnostic process challenging, mainly when ageing–related features occur [2]. OABD may be diagnosed later in life (LOBD) or as a relapse of a pre-existing condition (early onset, EOBD). OABD point prevalence rates were estimated to be between 0.1% to 0.4%, but a more significant proportion of older adults (about 10–25%) with a mood disorder may be diagnosed with BD [3,4]; therefore, the relative frequency of OABD has increased from 1% to 11% in recent decades [5]. OABD is frequently under-diagnosed, under-recognized, and not adequately treated [2], and methodological and clinical issues make this topic very challenging. Furthermore, intriguing new frontiers in the field of geriatric psycho-oncology are emerging thanks to several pieces of evidence supporting the biological link between mental illnesses, ageing, and cancer. Combined efforts to get proof to inform clinicians and policymakers are mandatory [6,7].

In 2019, a task force from the International Society for Bipolar Disorders (ISBD) published the methods and initial start-up activities for building an integrated OABD-focused database, the Global Aging and Geriatric Experiments in Bipolar Disorder Database (GAGE-BD) [8]. In our country, other studies explored the epidemiological framework of mental health disorders in the past [9] and recently [10,11], but with several limitations for the topic we addressed in this study. This article provides an overview of the current literature in this field, focusing on the Italian OABD epidemiological framework to suggest a potential area of improvement starting from pitfalls and challenges.

## 2. Materials and Methods

This study was performed at the Geriatric Unit based on AUSL-IRCCS of Reggio Emilia, with the participation of Executive Board members of the Italian Psychogeriatric Association (AIP)-Emilia Romagna section (Emilia Romagna is one of the biggest Italian regions in the North of Italy). AIP is an Italian scientific society that deals with neurodegeneration and psycho-behavioural disorders in the elderly, promoting research, education, and training in this field, and each Italian region has its own Executive Board representing neurologists, geriatricians, and psychiatrists [12].

We started with an exploratory literature review to identify the spectrum of research topics in this field and inferencing criticisms and challenges. Original and review articles were selected using the following keywords “bipolar disorders” AND/OR “mania” AND “elderly” AND/OR “geriatric population”. We searched Medline and PsycINFO for English-language articles published between January 2017 and December 2022 using MeSH headings. We identified some additional articles from references of relevant articles by hand. We screened our research, including studies with an age cut-off of 60 years. A Delphy-like approach was used to reach a consensus about thematic issues merged by reviewing articles. To investigate the national framework, we consulted the Italian public database [13] for mental health and report updated to 2021 based on the New Health Information System (Italian acronym NSIS). This system is an evolutionary step from the previous Informatics administrative flows as it collects data through various information sources to provide an overall picture of the activities of users of mental health services in Italy [14]. NSIS represents the reference tool for the quality, efficiency, and appropriateness of the National Health Service through the availability of information supporting regions and the Ministry of Health for policy-making decisions due to its completeness, consistency and timeliness. According to a Ministerial decree regulating the NSIS system, the classification diseases code still follows the Italian version of ICD-9 [15].

In this study, we analysed the cumulative data from the total psychiatric activities according to the current mental healthcare system. From the supplied Institutional summary table [Table 1], we extrapolated data regarding code 02 Mania and bipolar affective disorders and focused the analyses and data interpretation in the age range of 65–74 and over 75 years old.

In Italy, 75 years old is the age cut-off for considering people in the geriatric population [15]. According to a recent paper showing the European framework of psychiatric burden in older people, we set the superior age range at 84 years old; hence, we divided our population target into two groups, from 65 to 74 years and over 75 years old. Data were classified by gender and geographical distribution. A total of 22,937 men (13,358 and 9579 in the range of age 65–74 and 75–84 years, respectively) out of a total male population of 129,252 older than 18 years and 26,661 females (20,145 and 16,519 in the range of age 65–74 and 75–84 years, respectively) out of a total female population of 160,619.

Our variables of interest were: (1) Prevalence treated (the total number of patients referred to psychiatric services during the 2021 year with the mental health facilities) for 10,000 inhabitants; (2) incidence treated (number of new patients who have had their first-time contact with psychiatric facilities; and (3) finally, the “incidence treated–first ever” that we assumed as a proxy of LOBD with the severity of the disease. Patient refers to primary care first; it follows that a first-ever appointment directly in a specialized mental health service without previous contact with, i.e., the general practitioner, could be interpreted as a more severe case. All data were anonymous and authorised as these data are routinely collected for administrative purposes and converge in the NSIS of the Italian Ministry of Health. As the use of these data is mandatory to track healthcare performance for the Italian Healthcare System, the current regulation does not require the written consent of patients.

## 3. Results

We found various reports, from naturalistic observations of people (case reports, n: 225) to randomised controlled trials (n: 83); review articles (n: 43); and systematic reviews (n: 31) with meta-analyses (n: 28) observing a clinical heterogeneity BD across the life span. All case reports analysed reported the main thematic issues linked to managing comorbidities and adverse drug reactions in older people. Our research included only 23 studies for further interpretation according to our purpose. A lack of homogeneity came from the misleading definition in this field (older adults, older age, geriatric). The minimum age used to define OABD is generally 60 years [5,6]; however, some authorities use an age cut-off of 50, 55, or 65 [7]. The International Society for Bipolar Disorders (ISBD) Task Force uses the term “older age bipolar disorder” instead of “geriatric bipolar disorder”, recommending including patients aged ≥ 50 years [6]. Despite this, when we searched for late onset in the literature, we found an age cut-off of 40 years. There are no lifelong prospective studies on mood disorders, and data from extensive epidemiological studies reported substantially higher prevalence rates for both major depressive and bipolar disorders compared to the past. As polypharmacy is a concern in the elderly, the overlap between drug-induced mood disorders and chronic disease management can make the decision-making process confusing. Furthermore, there is limited evidence guiding treatments and non-pharmacological approaches in this target population due, for example to the stigma of referring to psychotherapeutic treatment, even amongst physicians. Methodological issues related to study design and assessment tools contribute to the confusing framework, so the lack of service development focused on older adults with BD and specific academic training could be affected by multi-level issues [Table 2].

In Table 3, we reported the recent projects specifically designed for the geriatric population.

The Italian network of healthcare services for mental disorders is made up of territorial, residential, and semi-residential facilities [Figure 1 and Table 4].

In 2021, the mental healthcare system involved 1245 Community mental health centres, and 1983 and 742 residential and semi-residential structures, respectively. Across the Italian regions, Calabria did not contribute to the administrative flow, and Sardinia failed to provide data regarding territorial and semi-residential from Local Health Agency located in their territory, whereas the others transmitted all records needed for analyses. The global prevalence of males treated ranged from 0.1% of Lombardia to 0.5% of the Autonomous Province of Bolzano and from 0.002% of Lombardia to 0.3% of the Autonomous Province of Trento for the group 65–74 years and 75+, respectively. For females, the prevalence rate ranged from 0.11% from Valle d’Aosta to 0.81% of the Autonomous Province of Bolzano and from 0.02% of Lombardia and 0.32% of the Autonomous Province of Trento for the group 65–74 years and 75+, respectively [Table 5]. We observed the same trend regarding the incidence; generally speaking, despite the similar prevalence rate across the whole country for both the age range subgroup, the incidence was slightly higher in the regions of the centre and southern Italy, especially for females in the range of 65–74 years [Table 6].

Table 7 shows data of OABD handled by the specialised service for the first time in their life (first ever) without previous contacts with others (i.e., GP). The Autonomous Province of Trento showed a higher incidence for both gender compared to other regions, with a higher value for females in the group of 65–74 years old and a light increase for the group of 75+ in males; for the remaining regions, females showed a higher global trend.

The observed trend for males ranged from 0.004% of Friuli Venezia Giulia to 0.0082% and from 0.003% of Lombardia to 0.071% of Autonomous Province of Trento for groups 65–74 and 75+ years old, respectively. The observed trend for females showed a fluctuating rate from the minimum of 0.009% to 0.13% of Autonomous Province of Bolzano in the group of 65–74 years old, during a minimum of 0.002% of Friuli Venezia Giulia to 0.063% of Autonomous Province of Trento for the group 75+ years.

## 4. Discussion

To our knowledge, this is the first study exploring the Italian framework of OABD, also considering people older than 75 years old. By analysing data for 2021, we settled on a starting point for further studies that would like to explore the impact of the COVID-19 pandemic in this field, considering the burden of mental illnesses after this period. OABD remain a nosologic dilemma, diagnostic challenge, and neglected area of research for different outcomes; however, in recent decades, this topic is gaining relevant interest in the scientific and clinical community [8]. The aetiology, clinical presentation, and management of BD in older adults have specific features compared to younger and improving the awareness of these differences are essential for an effective treatment. OABD could affect the ability to self-perform day-to-day tasks and living autonomously. The overlapping with behavioural and psychological mood in the realm of neurodegenerative or infectious diseases (i.e., limbic encephalitis) makes this field very challenging for clinicians [18,19]. Depression mood could be the first significant episode in about half of all BD patients, and the time lag until their first manic episode could be of 15 years; therefore, it is conceivable that people at 70 could be misdiagnosed as having late-life depression [20]. One of the main goals of this study was to illustrate potential intervention areas which foster future research and investments [Table 2]. The research question lay in the assessment of older adults with severe mood episodes without response to treatment or with negative outcomes. This issue emerged from the scientific debate among expert panellists in the regional section of the Italian psychogeriatric association.

Differences between the current and lifetime prevalence may be due to the use of different screening instruments and thresholds applied (cut-off ranges from 40 to 60 years). Previous studies supported our data presented in the Results section. Worldwide, the 12-month prevalence rate of BD in older patients is ~0.1–0.5%, with a difference by settings with prevalence in females [21,22,23]. Despite this relatively low prevalence of OABD in the community, data from clinical settings suggest that OABD is a frequent diagnosis related to hospital admission reaching 17% of the elderly patients presenting at emergency facilities. At specialised services, such as psychogeriatric units, 4–8% of the inpatients resulted an OABD diagnosis [23]. Andreas et al., in the MentDis_ICF65+ survey, assessed the 1-year incidence rate of the most frequent mental disorders across six different European in community-dwelling elderly aged 65–84 years [17] reporting the 1-year incidence rates OABD in community-dwelling worldwide for the first time [17]. BD account for the current prevalence of around 8.0%, and this study showed that the burden of mental illness in the elderly was previously underestimated. The only Italian centre involved was Ferrara, a city based in Emilia Romagna, and its data are probably not representative of the whole country. Furthermore, we found that in the study limitations, the authors reported that Ferrara missed providing complete data due to the earthquake [17]. Another study by Silvestri and colleagues investigated the prevalence of mental disorders in a Northern Italian region [10]. The lifetime prevalence of bipolar I-II disorders was higher than the one reported by Faravelli et al. previously [24], and the point prevalence of OABD was 1.8% in people aged 75+ for BD II (*p* = 0.21, in the point prevalence by age groups). The authors underlined that hypomanic episodes during a lifetime could be challenging to detect by one interview assessment, and the prevalence of bipolar II disorder could have been underestimated.

Gender and cultural differences in psychiatric disorders may lead to underestimating psychiatric conditions in some categories. In our study, prevalence and incidence were higher in the female group and, in particular, in the group 65–74 years. Females are over-represented as users of mental health services, maybe due also to the males’ reluctance to speak about their mental health problems. Baby Boomers are a generational cohort currently consisting of adults in their early to mid-seventies. Baby Boomers generally take a sceptical and challenging approach toward mental health and seek mental health treatment [25]. Accurate diagnostic workflow, including a close follow-up to distinguish psychiatric from neurodegeneration diseases or alternative diagnosis, including drugs (corticosteroid-induced mania, sympathomimetic, or anti-Parkinson drugs), is necessary. While people with a historical relationship of psychiatric disorders can easily interpret symptoms under specific diagnostic categories (early onset, recurrence), in general, LOBD overlaps with other disorders typically related to ageing and having a suspicion of illness could be remarkable. If the first episode occurs at 70, the chance of being affected by other diseases (i.e., stroke or other neurodegenerative disorders or risk of all causes of mortality could modify the prevalence of this disorder [26]. For this, psychiatric and geriatric co-management is recommendable to solve ambiguous clinical features of BD in the elderly and fostering education and training in this field also for primary care.

The lack of specific services for the elderly with psychiatric disorders contributes to the underestimation. Considering that approximately 70% of older adults with prevalent mood disorders did not use mental health services for social stigma [25], the geriatric psychiatrist’s role could be crucial in detecting older people with mental illnesses, particularly OABD. The field of geriatric psychiatry is in demand, and it must be acknowledged that there are not enough geriatric psychiatrists to meet the needs of ageing [27]. Given the sheer scale of the elderly population, the number of geriatric psychiatrists is not currently sufficient to guarantee the standard of quality care that this population deserves. In Italy, the figure of a geriatric psychiatrist in the staff of mental healthcare services is still missed. Furthermore, the subspecialty training and certification in Geriatric Psychiatry is chosen voluntarily by residents and in mental healthcare facilities, geriatric psychiatrics are not required. With the ageing population, it is fair to assume that the portion of old-age patients suffering from BD will grow analogously. So, it is time to rethink the training system in medical school and primary care.

While the Department of Mental Health operating in the Italian territory aims to take care and rehabilitate adult people with psychiatric disorders (after discharge from the mental hospitals legally closed in 1980), the psychogeriatric unit usually refers to Community Mental Health Centres that provide care for elderly with neurodegenerative disease [28]. In Italy, the reform of the Mental healthcare system has not been updated since the 1990s nor adapted to demographic changes. We have to deal with two kinds of scenarios: (1) psychiatric people who are ageing and (2) elderly suffering from late-onset psychiatric morbidity. While the additional strain generated by the pandemic is new, the mental health situation and lack of services have been a neglected human rights crisis in Europe for a long time [29]. There is a call for governments to manage essential social determinants of health to rebuild more inclusive and resilient healthcare systems, prevalently for mental healthcare.

The Joint Commissioning Panel for Mental Health (JCP-MH) provided a real-world guide pointed to the need to organise healthcare services according to clinical needs and not by age alone [30]. Older people’s mental health services should not be incorporated into a broader ‘adult mental health’ or ‘ageless service’. The needs of older people with functional mental illness and/or organic diseases, such as dementia and their associated physical and social issues are often distinct from younger people, and some aspects deserve further consideration. First, the assessment phase: following the guidelines of the International Society for Bipolar Disorders-Battery for Assessment of Neurocognition (ISBD-BANC) [31], a neuropsychological battery exploring 12 cognitive tasks for 5 different cognitive domains lasts approximately 80 min. This time might be challenging for older people, and results may be inconclusive. Second, the therapeutic challenges when comorbidities are present. The GERI-BD project aimed to assess various clinical domains before and during pharmacological treatment in older adults [6]. This study showed how many challenges researchers studying drug treatments for OABD have to face in this category of patients. Therefore, there is a need to foster the research by deploying another digital approach, for example, machine learning and artificial intelligence-based algorithm.

Based on medical and epidemiological data, surveys, and meta-regression models, we observed an increased OABD trend overtime, particularly highest in Australasia, Tropical Latin America, high-income North America, and amongst Indigenous people in New Zealand [32,33], suggesting some genetic or environmental influence. In our study, the Autonomous Provinces of Bolzano and Trento detected more cases in their population than in other regions. Even though this could be biased by cultural differences or differences in the clinical care-pathway organisation due to the autonomy in the healthcare decisions, this is the first attempt in our country to give a global framework on OABD. Unfortunately, data about racial or other demographic characteristics were not retrievable for our study population, and migration’s effect on social and genetic issues is a theme worthy of being addressed with specific studies.

BD is associated with the risk of suicide in the young population [34], while suicide rates among older patients have not been explicitly studied, although the risk of suicide in old age has become a primary global concern for both public and mental health [10,35]. Our data allowed us to define the prevalence and incidence of OABD users treated that we could define as situations at risk. We explored the incidence of new treatment as a proxy of clinical severity, and our results’ overall direction showed trends worthy of being addressed in further research. Our results are the first attempts to shed light on future needs and steer healthcare policies to mitigate complex situations with multilevel strategies involving mental healthcare and the community.

Another intriguing field of research is the suggested biological link between severe mental disorders, including bipolar disorder, and malignant cancers in these last decades [36]. BD has been associated with clinical signs of accelerated ageing, potentially underlying its association with several age-related medical conditions, including cancer. Previous evidence demonstrated that patients with BD, both men and women, were more likely to develop cancer, approximately 20% higher than in the general population, with a higher incident risk for breast cancer in women with bipolar disorder [37]. A recent meta-analysis study of 4,910,661 BD individuals, including nine studies, showed an increased risk of all-cause cancer [relative risk (RR) = 1.24, 95% Confidence of Interval (CI) (1.05, 1.46)], in particular breast cancer in females [RR = 1.33, 95% CI (1.15, 1.55)] [38]. In a nationwide population-based study based in Taiwan, Chen et al. demonstrated that oral cavity cancers was more frequent in individuals aged <50 years with bipolar disorder [Odds Ratio (OR) = 2.16, 95% CI (1.36, 3.30)]; however, no increased risk of cancer development was observed during the follow-up period for those aged ≥ 50 years [36]. Lithium is the principal drug used for BD prevalently in young people, so the relationship between BD and cancer could be drug-mediated or related to some pharmacodynamic properties [37]. A systematic review with a random-effects model (59,000 patients with BD and 4500 individuals treated with lithium) revealed that people with BD had an increased risk of developing cancer but no significant relationship with lithium [38]. What is more, the results of the study point to lithium treatment having a possible protective effect of around 5% against the risk of developing cancer.

The reasons contributing to the higher cancer prevalence in patients with BD are varied and complex, including socio-demographics, lifestyle factors, especially unhealthy behaviours, hormonal, immune, and inflammatory changes, which lead to “accelerated ageing” and a genetic and epigenetic predisposition [39]. Age is one of the main risk factors for cancer. Phenotypically, ageing differs from person to person as we distinguish fit, pre-frail and frail people, not only by age. As our population is progressively ageing, the proportion of older patients with cancer is increasing significantly. Evidence suggested that some medications used to treat BD may be protective against accelerated ageing effects; in fact, a positive correlation between the most frequently used mood stabilise (lithium), and its duration treatment and telomere length has been demonstrated [40,41,42]; furthermore, older patients with BD seem to present higher levels of growth differentiation factor 15, a biomarker of ageing associated with a longer length of illness [42]. This supports the need to rule out BD from other mental conditions, avoiding underdiagnosis and providing appropriate treatments. Promoting further research in this perspective is worthy of the global concern about the ageing population, cancer, and mental health.

## 5. Limitation

It must be taken into account that a share of people with psychiatric distress do not turn to specialist services but are partially treated in family medicine or within the private sphere and partly remains outside the health system, so our results are likely underestimated, and this assumption is in line with the current literature in this field. Our study found some regional differences in the OABD prevalence and incidence, and despite the intriguing perspective of somewhat environmental and genetic traits influencing psychiatric disorders, our results could be more realistic attributable to organisational and resource-dependent factors. As the principal purpose of this study was to have a snapshot from the last 5 years on this topic as a substrate for the interpretation of our national framework, we did not perform an extensive literature review, but we only explored the research question under the literature review methodology with selected filters. This could be a limitation to the final and conclusive suggested solutions to the OABD in the clinical and research field. Furthermore, some sites did not provide data and the motivation for missing them was not documentable. However, we also referred to these sites to provide a comprehensive framework for our country in the hope of stimulating constructive debate and aligning regions with ministerial duties.

## 6. Conclusions

Bipolar disorders in the elderly are increasingly recognised as an affective disorder with specific characteristics. The number of elderly people affected is increasing, with more people with chronic disorder reaching older ages, and meanwhile, bipolar disorders at their first onset in geriatric age challenging our mental health care system. There is an increasing need for deeper knowledge in this area, such as physiological changes associated with ageing and comorbidity, which make these individuals more prone to side effects and drug interactions and to the risk of not being properly diagnosed. In particular, OABD has a high rate of neurological comorbidities and cardiovascular risk factors and an overlapping of clinical symptoms, so a comprehensive approach to address these clinical-related problems through close coordination with primary care is compelling as well the improvement in education and training programs at medical schools. Gender differences in older age need to be better explored in their underlying bio-psychosocial mechanisms so as to drive clinical and policy-making decisions. Over the past three decades, more work has been done to better understand the disease in this age group and create research consortia. OABD is at the crossroads of several fields of research, including geriatrics, psychiatry, neuroscience, immunosenescence, and cancer risk. Researchers should address still open issues and promote fundamental research in neuroscience and geriatric psycho-oncology. Significant changes in clinical practice could result from increasing attention to this problem. Providing an overview of the epidemiology of OABD by establishing the appropriate age threshold to define the geriatric population in our country is essential. This was the first attempt to shed light on the Italian mental healthcare system to solicit debate about resource allocation and clinical needs and to fix a starting point for further considerations. Potential consequences of OADB on cancer risk must be addressed by increasing the awareness of family physicians, geriatricians, and psychiatrists and by adapting screening programs so that early stage forms can be diagnosed. Contributing to the research in this field, joining efforts within the European consortium is a fascinating proposition that we will develop within future AIP initiatives.

## Figures and Tables

**Figure 1 biomedicines-11-01418-f001:**
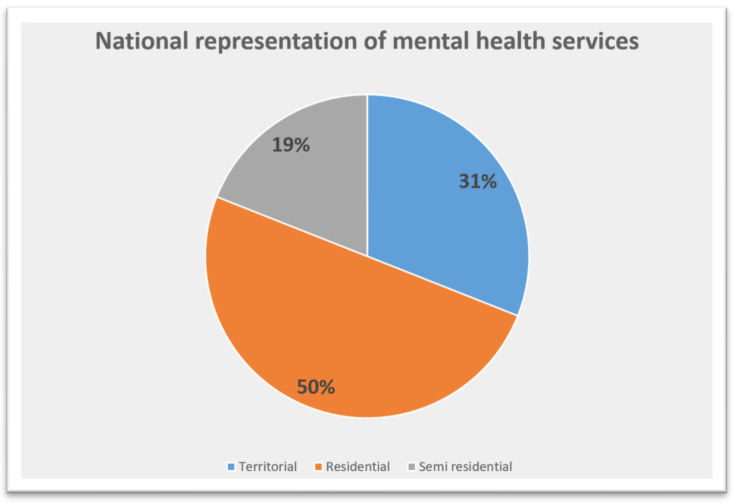
Distribution of mental health care services in Italy (%).

**Table 1 biomedicines-11-01418-t001:** Summary and code for mental health diagnostic groups. Adapted from the Italian Report on Mental Illnesses [10].

DIAGNOSTIC GROUPS	ICD9-CM CODE
01–Schizophrenia and Other Functional Psychosis	295, 297, 298 (EXCEPT 298.0), 299
02–Mania and Bipolar Affective Disorders	296.0, 296.1, 296.4-8
03–Depression	296.2-3, 296.9, 298.0, 300.4, 309.0, 309.1, 311
04–Nevrotic and Liason Syndromes	300 (except 300.4), 306, 307.4, 307.8-307.9, 308, 316
05–Personality and Behavioral Disorders	301, 302, 312
06–Alcohol and Substance Addictions	291, 292, 303, 304, 305
07–Dementia and Organic Mental Disorders	290, 293, 294, 310, 293, 294
08–Mental Retardation	317, 318, 319
09–Other Psychiatric Disorders	307.0-307.3, 307.5-307.7, 309.2-309.9, 313, 314, 315

**Table 2 biomedicines-11-01418-t002:** Principal issues and suggestions for intervention areas for Bipolar Disorders in the elderly field.

Principal Issues	Suggestion for Intervention Area
Evidence base guiding treatment remains limited.	Pharmacological studies, specifically conducted in older patients
Clinical heterogeneity in BD across the lifespan	Personalized approach, implementing predictive model by Machine learning
BD is under-recognized in primary care	Improvement of education and training in primary care, pursuit of knowledge web platform, increase awareness
Lack of specific and standardised assessment tools	Meeting challenges to assessment
Treatments for medical disorders may contribute to mood disorder symptoms, and some mood disorder treatments may contribute to or exacerbate medical conditions	Knowledge–based pharmacological interactions match the physiological role of ageing, cognition, function, and physical health.
Overlapping between psychiatric features and cognitive impairment	Foster investigation in the neurobiology of BD in the elderly (i.e., by artificial intelligence techniques)
Poor evidence from an uncontrolled, open label, or exploratory trials of larger mixed-age populations studies	Meeting challenges for RCT in the elderly
Limited evidence of non-pharmacological approach in this target population	Promoting research projects in this field
Lack of service development focused on older adults with BD and specific academic training	Promoting debate with stakeholders and policy-makers

**Table 3 biomedicines-11-01418-t003:** Initiatives to advance the knowledge of OABD.

Authors	Year	Project	Description
Marino et al. [16]	2017	The NIH-funded multicenter study Acute Pharmacotherapy of Late-Life Mania (GERI-BD)	Assessing various clinical domains before and during mood stabiliser treatment in older adults participating in a 9-week, double-blind, randomized controlled trial focus on BD type I
Sajatovic et al. [8]	2019	The Global Aging and Geriatric Experiments in Bipolar Disorder Database (GAGE-BD)	Large integrated sample comprising archival studies about OABD
Andreas et al. [17]	2022	The European MentDis_ICF65+ study	Cross-sectional and longitudinal multi-centre survey to collect data on the prevalence and incidence of any mental disorder in the European elderly

**Table 4 biomedicines-11-01418-t004:** Italian mental health services: regional distribution.

REGION	TERRITORIAL FACILITIES	RESIDENTIAL FACILITIES	SEMI-RESIDENTIAL FACILITIES
PIEMONTE	84	352	43
VALLE D’AOSTA	6	6	2
LOMBARDIA	152	317	139
PA BOLZANO	11	10	4
PA TRENTO	10	13	6
VENETO	203	232	113
FRIULI VENEZIA GIULIA	22	26	11
LIGURIA	25	74	20
EMILIA ROMAGNA	57	120	28
TOSCANA	237	138	111
UMBRIA	18	66	23
MARCHE	34	71	22
LAZIO	79	121	52
ABRUZZO	16	34	16
MOLISE	3	9	1
CAMPANIA	67	34	46
PUGLIA	45	236	51
BASILICATA	9	25	5
CALABRIA	nd	nd	Nd
SICILIA	143	99	49
SARDEGNA	24	-	-
ITALIA	1245	1983	742

**Table 5 biomedicines-11-01418-t005:** Prevalence of bipolar disorders by gender in the elderly: regional estimates.

REGION	MALES		FEMALES	
	65–74	75+	65–74	75+
PIEMONTE	0.20	0.07	0.23	0.07
VALLE D’AOSTA	0.15	0.04	0.11	0.07
LOMBARDIA	0.10	0.02	0.13	0.02
PA BOLZANO	0.50	0.22	0.81	0.31
PA TRENTO	0.32	0.28	0.52	0.32
VENETO	0.15	0.07	0.22	0.09
FRIULI VENEZIA GIULIA	0.14	0.05	0.22	0.08
LIGURIA	0.29	0.11	0.37	0.12
EMILIA ROMAGNA	0.17	0.07	0.24	0.09
TOSCANA	0.13	0.06	0.20	0.07
UMBRIA	0.27	0.09	0.45	0.21
MARCHE	0.14	0.06	0.21	0.06
LAZIO	0.12	0.05	0.18	0.06
ABRUZZO	0.11	0.06	0.14	0.04
MOLISE	0.21	0.05	0.25	0.11
CAMPANIA	0.12	0.04	0.15	0.04
PUGLIA	0.14	0.04	0.14	0.04
BASILICATA	0.24	0.06	0.27	0.08
CALABRIA	-	-	-	-
SICILIA	0.11	0.04	0.13	0.04
SARDEGNA	0.16	0.04	0.27	0.07

**Table 6 biomedicines-11-01418-t006:** Incidence of bipolar disorders by gender in the elderly: regional estimates.

REGION	MALES		FEMALES	
	65–74	75+	65–74	75+
PIEMONTE	0.009	0.004	0.016	0.005
VALLE D’AOSTA	0.013	0.015	-	0.030
LOMBARDIA	0.018	0.003	0.02	0.006
PA BOLZANO	0.074	0.048	0.14	0.060
PA TRENTO	0.082	0.071	0.11	0.063
VENETO	0.041	0.017	0.048	0.021
FRIULI VENEZIA GIULIA	0.004	0.006	0.009	0.002
LIGURIA	0.054	0.025	0.043	0.015
EMILIA ROMAGNA	0.031	0.0177	0.044	0.016
TOSCANA	0.033	0.016	0.052	0.015
UMBRIA	0.031	0.004	0.046	0.016
MARCHE	0.006	0.006	0.006	0.002
LAZIO	0.052	0.026	0.085	0.036
ABRUZZO	0.058	0.034	0.079	0.022
MOLISE	0.048	0.024	0.033	0.024
CAMPANIA	0.04	0.018	0.048	0.016
PUGLIA	0.043	0.012	0.041	0.011
BASILICATA	0.015	0.007	0.017	-
CALABRIA	-	-	-	-
SICILIA	0.025	0.009	0.026	0.011
SARDEGNA	0.015	0.005	0.019	0.006

**Table 7 biomedicines-11-01418-t007:** Incidence of first-ever treatment for Bipolar Disorder in the Italian geriatric population: regional estimates.

REGION	MALES		FEMALES	
	65–74	75+	65–74	75+
PIEMONTE	0.007	0.004	0.014	0.005
VALLE D’AOSTA	0.013	0.015	-	0.030
LOMBARDIA	0.017	0.003	0.020	0.006
PA BOLZANO	0.066	0.040	0.13	0.060
PA TRENTO	0.082	0.071	0.11	0.063
VENETO	0.039	0.017	0.044	0.020
FRIULI VENEZIA GIULIA	0.004	0.006	0.009	0.002
LIGURIA	0.044	0.022	0.038	0.013
EMILIA ROMAGNA	0.029	0.016	0.039	0.014
TOSCANA	0.032	0.016	0.049	0.015
UMBRIA	0.031	0.004	0.039	0.016
MARCHE	0.006	0.006	0.005	0.002
LAZIO	0.045	0.023	0.073	0.033
ABRUZZO	0.054	0.028	0.069	0.022
MOLISE	0.048	0.018	0.025	0.024
CAMPANIA	0.039	0.017	0.044	0.016
PUGLIA	0.042	0.012	0.041	0.011
BASILICATA	0.015	0.007	0.017	-
CALABRIA	-	-	-	-
SICILIA	0.023	0.008	0.023	0.010
SARDEGNA	0.014	0.005	0.018	0.006

## Data Availability

Data are available upon direct request to the corresponding author.

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
