# Peer review of "The Italian Framework of Bipolar Disorders in the Elderly: Old and Current Issues and New Suggestions for the Geriatric Psycho-Oncology Research"

_biomedicines, 2023, doi:10.3390/biomedicines11051418_

Round 1
Reviewer 1 Report
Thanks for recommending me as a reviewer. In this paper, the authors aimed to describe the state-of-the-art of OABD in an Italian framework and provide a new research field. The authors performed an overview of the literature to select a target population (65 years and older) and synthesize key issues. The authors analyzed epidemiological data in the 65–74 and 75–84 age groups utilizing the 2021 Ministry of Health Italian database. In this paper, women had the highest prevalence and incidence in both groups, with regional differences across the country, but more evident in the municipalities of Bolzano and Trento in the 65–74 year age range. If the authors complete minor revisions, the quality of the study will be further improved.
1. The introduction section is well written. However, some paragraphs consisted of one sentence. A paragraph cannot consist of a single sentence.
2. Is there a reason for analyzing with ICD-9 rather than ICD-10 in Table 1?
3. In Figure 1, the pie chart and table should be separated.
4. Shouldn't figures 2,3,4 be presented as tables?
5. In the conclusion section, the authors should emphasize the key findings of this study. The conclusion section needs to be rewritten
Author Response
On behalf of my colleagues and me, I would like to thank the reviewer the time ad consideration of our manuscript. Overall, thanks for his/her/* suggestions.
The introduction section is well written. However, some paragraphs consisted of one sentence. A paragraph cannot consist of a single sentence.: we changed accordingly
2. Is there a reason for analyzing with ICD-9 rather than ICD-10 in Table 1?. As we stated in the text ( Line 96-101) the national informative system by ministeral decree was set up on the ICD 9 italian version 2002 and there is not an update of the administrative flow. To be coherent with the database and subsequent results we left this classification disease version.
3. In Figure 1, the pie chart and table should be separated.: thank you for your suggestion, we provided the needed split.
4. Shouldn't figures 2,3,4 be presented as tables? Addressed, we changed them in tables and amended the text.
5. In the conclusion section, the authors should emphasize the key findings of this study. The conclusion section needs to be rewritten.
Thank you for this important suggestion, we rewrote the conclusion section. We were careful with words count but if in the reviewer's opinion we can improve more we are available to work on it.
Reviewer 2 Report
Title: should not have full stop after end of the title, so please remove it. is it old issues or current issues?
Introduction: please combine 2nd, 3rd, and 4th paragraphs into one paragraph.
Line 56-57: "OABD prevalence rates were estimated to be between 0.1% to 0.4%, 10% to 56 25%", first % between 0.1% to 0.4% and second % data is between 10% to 25%, these numbers have large discrepancies. Are these two numbers different prevalence rate?
Methods: Please mention how many females and males' participants were considered in this study and their age ranges.
Line 81: Please rephrase "to get a general picture"
Line 139: please combine this statement to previous section and elaborate about the table 2 in the text.
Table 2: pitfalls-- current issues or old issues? is Intervention are suggestion?
Line 159-162: "For females, the prevalence 159 ranged from 0.11 from Valle d’Aosta to 0.81% of the Autonomous Province of Bolzano 160 and from 0.02% of Lombardia and 0.32% of the Autonomous Province of Trento for the 161 group 65-74 ys and 75+ respectively [Fig.2]." Please rephrase this sentence. Are these numbers prevalence rate? if so please mention how you calculated these numbers.
Figure 2: Please change from figure to table.
Figure 3: Please change from figure to table.
Figure 4: Please change from figure to table.
Line 251: word in is repeated. Please correct it.
Line 254: Please cite this sentence ('The number of psychiatrists caring for the older population needs to double') if it is a true statement or if it is general statement then please remove or write like more psychiatrists required.
Lin 226: please change from in our country to In Italy.
Line 268: open bracket has to closed.
Line 270: remove 'a' from: "have been a neglected human"
Line 279: "Several points are worthy of being addressed. First of all, the assessment phase": Please elaborate which points to worth to address?
General Suggestion: Too many short sentences throughout the manuscript, so please rephrase them for readers to have some flow while reading the manuscript.
Line 317: "4 910 661", if this is one number then remove space and add comma.
Line 317-318: What these number (1.24 and 1.13 and) represent? Are they %?
Line 321: Remove double "."
Line 356: change "so" to "thus"
Strengths are generally discussed in the beginning of your discussion section and provide concise context in the conclusion section. Thus, please removes special Strengths in discussion section. Elaborate the limitations of your study, for example why you only selected fewer studies and sites that are not providing data etc.
Comments on quality of English are given in the earlier section.
Author Response
On behalf of my colleagues and me, I would thank the reviewer for her/his/* comments, time and consideration of our manuscript.
We provided our reply point by points as follows ( maybe in the new version some lines are not correspondent)
Title: should not have full stop after end of the title, so please remove it. is it old issues or current issues?
Thanks for this, I admit it was an oversight and I apologize for this: Removed.
- Introduction: please combine 2nd, 3rd, and 4th paragraphs into one paragraph. : addressed, thanks.
- Line 56-57: "OABD prevalence rates were estimated to be between 0.1% to 0.4%, 10% to 56 25%", first % between 0.1% to 0.4% and second % data is between 10% to 25%, these numbers have large discrepancies. Are these two numbers different prevalence rate? We reformulated the sentence to make the statement more understandable as the first percentage is related to the OABD point prevalence from literature before the expert consensus meeting, while a percentage up to 25% is related to the umbrella term of mood disorders where according with the recent evidence these disorders could mask a BD. Hope in this way the meaning could be better: “While OABD point prevalence rates were estimated to be between 0.1% to 0.4%, a more significant proportion of older adults (about 10-25%) with mood disorder may be diagnosed with BD according with some recent evidence [3, 4]; therefore, the relative frequency of OABD has increased from 1% to 11% in recent decades [3, 4]. We apologize if the reviewer meant something else and we are going to be available to modify again to take up the reviewer’s suggestion.
- Methods: Please mention how many females and males' participants were considered in this study and their age ranges.: Thank you for this fruitful comment, we specified this aspect: (please, see line 109-112)
- Line 81: Please rephrase "to get a general picture" :
- Line 139: please combine this statement to previous section and elaborate about the table 2 in the text. Thank you for this comment, we modified accordingly.
- Table 2: pitfalls-- current issues or old issues? is Intervention are suggestion? Thanks you for your comment and the opportunity to think about this. We agree and we modified accordingly.
- Line 159-162: "For females, the prevalence 159 ranged from 0.11 from Valle d’Aosta to 0.81% of the Autonomous Province of Bolzano 160 and from 0.02% of Lombardia and 0.32% of the Autonomous Province of Trento for the 161 group 65-74 ys and 75+ respectively [Fig.2]." Please rephrase this sentence. Are these numbers prevalence rate? We rephrase the sentence specifying that our data refer to the prevalence rate. We calculated the proportion of a population who have a BD diagnosis on the whole population affected by mood disorders in that area for the period considered.
- Figure 2: Please change from figure to table. : change made
- Figure 3: Please change from figure to table. Change made
- Figure 4: Please change from figure to table. Change made
- Line 251: word in is repeated. Please correct it: correction made
- Line 254: Please cite this sentence ('The number of psychiatrists caring for the older population needs to double') if it is a true statement or if it is general statement then please remove or write like more psychiatrists required. We agree with this sugegstion and modified accordingly.
- Lin 226: please change from in our country to In Italy.: addressed
- Line 268: open bracket has to closed. We rephrased as follows: We have to deal with two kinds of scenarios: 1) psychiatric people who are ageing and 2) elderly suffering from late-onset psychiatric morbidity.
- Line 270: remove 'a' from: "have been a neglected human": removed
- Line 279: "Several points are worthy of being addressed. First of all, the assessment phase": Please elaborate which points to worth to address? Maybe the previous sentence could be misunderstood . We meant that some aspects deserve further considerations. First of all…..Hope I could make the point that needed to be made and I apologise if I ‘m still wrong
- General Suggestion: Too many short sentences throughout the manuscript, so please rephrase them for readers to have some flow while reading the manuscript.: Thank you we provided a new version to make the reading flowing
- Line 317: "4 910 661", if this is one number then remove space and add comma.
- Line 317-318: What these number (1.24 and 1.13 and) represent? Are they %? They represent the relative risk to develop cancer disease for people affected by BD. This is reference not from the results of our study but from the reference 38.
- Line 321: Remove double ".": addressed
- Line 356: change "so" to "thus"; addressed
- Strengths are generally discussed in the beginning of your discussion section and provide concise context in the conclusion section. Thus, please removes special Strengths in discussion section. Elaborate the limitations of your study, for example why you only selected fewer studies and sites that are not providing data etc. Thank you for your suggestion and we modify accordingly and I hope I was able to make our manuscript improved in this direction.

Round 2
Reviewer 2 Report
Thanks for address all of my comments and suggestions. One more minor suggestion please make sure you use word table tool to create the tables. The present form of tables is not in the acceptable forms.